# Correlation of optic nerve and optic nerve sheath diameter with intracranial pressure in pigs

R. Mija[1], I. Zubak[1], A. Schuetz[1], M. Glas[2], C. Fung[1], S. M. Jakob[2], J. Beck[1], W. J. Z'Graggen[1], Andreas Bloch[2]*

1 Department of Neurosurgery, Inselspital, Bern University Hospital, University of Bern, Bern, Switzerland, 2 Department of Intensive Care Medicine, Inselspital, Bern University Hospital, University of Bern, Bern, Switzerland

* andreas.bloch@insel.ch

## Abstract

### Objective

Several studies have shown an association between intracranial pressure and the diameter of the optic nerve sheath measured by transbulbar ultrasonography. To understand the pathophysiology of this phenomenon, we aimed to measure the changes of the optic nerve, optic nerve sheath and perineural space separately with increasing intracranial pressure in a porcine model.

### Methods

An external ventricular drain was placed into the third ventricle through a right paramedian burrhole in eight anesthesized pigs. The diameters of the optic nerve and the optic nerve sheath were measured while the intracranial pressure (ICP) was increased in steps of 10mmHg from baseline up to 60 mmHg.

### Results

*The median diameters of the optic nerve (ON) increased from 0.36 cm (baseline– 95% confidence interval (CI) 0.33 cm to 0.45 cm) to 0.68 cm (95% CI 0.57 cm to 0.82 cm) at ICP of 60 mmHg (p<0.0001) and optic nerve sheath (ONS) from 0.88 cm (95% CI 0.79 cm to 0.98 cm) to 1.24 cm (95% CI 1.02 cm to 1.38 cm) (p< 0.002) while the median diameter of the perineural space (PNS) (baseline diameter 95% CI 0.40 cm to 0.59 cm to diameters at ICP 60 95% CI 0.38 cm to 0.62 cm) did not change significantly (p = 0.399). Multiple comparisons allowed differentiation between baseline and values ≥40 mmHg for ON (p = 0.017) and between baseline and values ≥ 50mmHg for ONS (p = 0.006). A linear correlation between ON ($R^2$ = 0.513, p<0.0001) and ONS ($R^2$ = 0.364, p<0.0001) with ICP was found. The median coefficient of variation for intra- and inter-investigator variability was 8% respectively 2.3%.*

**Data Availability Statement:** All relevant data are within the paper and its Supporting Information files.

**Funding:** AB received a restricted grant from the foundation for research in anesthesia and intensive care medicine, Inselspital, Bern University Hospital, University of Bern, Switzerland. Grant number: 24/2016. URL: https://stiftungschweiz.ch/organisation/stiftung-fuer-die-forschung-in-anaesthesiologie-und-intensivmedizin The funders had no role in study design, data collection and analysis, decision to publish, or preparation of the manuscript.

**Competing interests:** The authors have declared that no competing interests exist.

## Conclusion

Unexpectedly, the increase in ONS diameter with increasing ICP is exclusively related to the increase of the diameter of the ON. Further studies should explore the reasons for this behaviour.

## Introduction

To obtain accurate transbulbar ultrasonographic measurements the investigator must be familiar with the anatomical structures inside the orbit. Both the optic nerve (ON) and the optic nerve sheath (ONS) appear as hypoechogenic structures. The ON is separated from the ONS by the perineural space (PNS) which is a hyperechogenic layer surrounding the ON (Fig 1) [1].

Several studies have shown that the diameter of the optic nerve sheath (ONS) correlates with changes in intracranial pressure (ICP) [2–4]. In critically ill patients, relative but not absolute ICP changes can be monitored bedside using ultrasound [5–7]. Hansen et al. studied ONS changes in patients undergoing lumbar infusion tests when cerebrospinal fluid (CSF) pressure was gradually raised. This was associated with a linear increase in ONS [8]. In a porcine model, a balloon catheter in the superior vena cava led to elevated cephalic venous pressure with consecutive ICP increase which could be tracked by increasing ONS diameter [9]. Further dynamic ONS assessment has been used in the diagnosis of spontaneous intracranial hypotension by determining the respective change between supine and upright body position [10].

So far, studies have not differentiated changes in diameter of ON and PNS with increasing ICP. Since PNS is a subarachnoid cistern, one would assume that acute ICP changes expand this CSF filled space. On the other hand, capillaries and venules within ON may become congested with increasing ICP and ON diameter may therefore increase as well. The aim of our study is to characterize the individual behaviour of ON and PNS with increasing ICP levels. We hypothesized that both ON and PNS diameter increase with increasing ICP.

Further we aimed to assess the accuracy and the reproducibility of transbulbar ultrasonography by measuring the intra- and inter-investigator variability.

## Materials and methods

The study was performed in accordance with the Swiss national guidelines for the Care and Use of Laboratory Animals, National Academy of Sciences (1996). An approval of the Commission of Animal Experimentation of the Canton Bern, Switzerland was obtained (approval number BE 40/16). This manuscript adheres to the applicable ARRIVE guidelines.

In compliance with the three R principle, the same group of pigs were subject to three different, unrelated studies, all involving ultrasound. First, a study investigating muscle compartment pressure in both lower legs was conducted [11]. After a stabilisation phase and establishment of a second baseline, a study on abdominal pressure was obtained [12]. Finally, this present study was performed.

### Animal preparation

Pigs were chosen because of their size which allowed instrumentation similar to that in humans. Animal preparation was described in detail previously [11–13]. In brief, 8 domestic pigs (weight 39±2kg (mean±SD)) were anesthetized using 20mg/kg ketamine and 2mg/kg

## baseline ICP

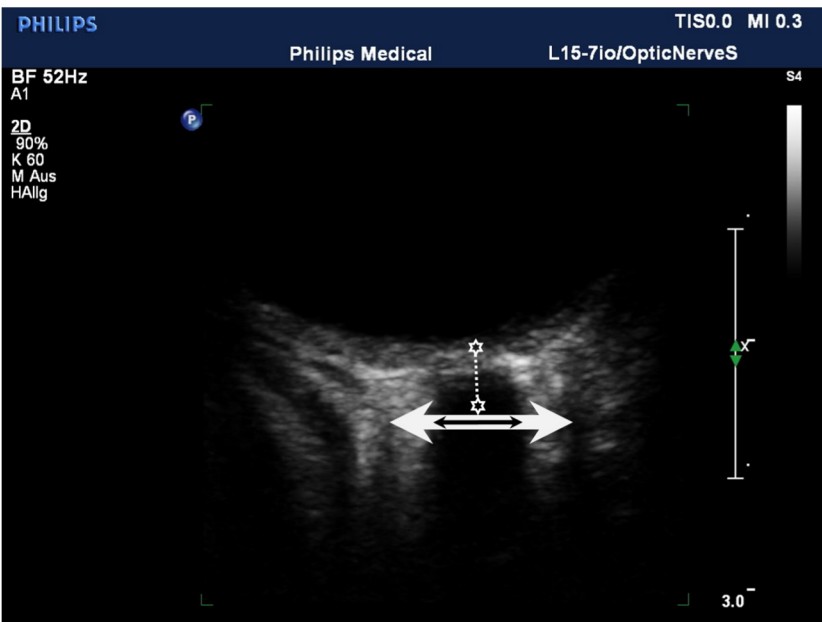

## ICP 40 mmHg

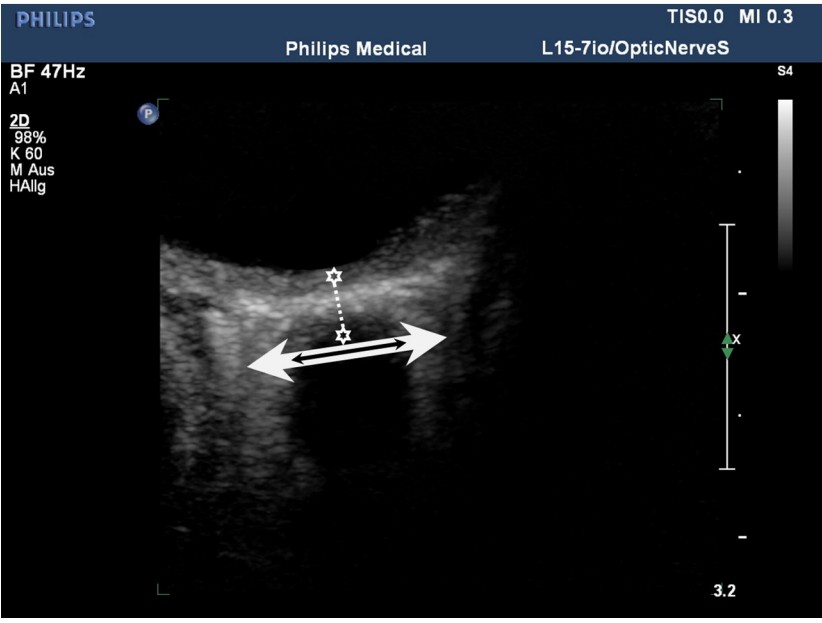

**Fig 1. The upper transbulbar ultrasound image was recorded at baseline ICP–the lower image was recorded at ICP of 40mmHg.** All measurements were obtained 3 mm behind the papilla as indicated by the two stars connected with the dotted line. Optic nerve diameter increased from 0.36 cm to 0.48 cm as illustrated by the black arrow. Optic nerve sheath diameter increased from 0.79 cm to 0.86 cm as illustrated by the white arrow. In both ultrasound images the lens was bypassed in order to obtain the most accurate display of the posterior structures of the eye. Further both images are made by using a zoom box of 2 cm size.

xylazine intramuscularly. Midazolam (0.5mg/kg) and atropine (0.02mg/kg) were administrated and the animals were orally intubated. An intravascular catheter was placed in the left internal carotid artery in order to invasively monitor blood pressure.

At the end of the experiment a bolus of 40 mmol potassium chloride was given to sacrifice the animal.

## Hemodynamic monitoring

Pulse oximetry and arterial pressure (MAP) were recorded with pressure transducers (xtrans®, Codan Medical, Germany). The recorded values were displayed continuously on a multi-modular monitor (S/5 Critical Care Monitor®; Datex-Ohmeda, GE Healthcare, Helsinki, Finland).

## External ventricular drain placement

The pigs were positioned in a left lateral decubitus position for all subsequent study interventions and measurements. A right-sided external ventricular drain catheter with a bolt-kit system for fixation and an air-pouch balloon situated at the tip for ICP measurement (Silverline-Bolt-Kit-catheter®; Spiegelberg GmbH&Co.KG; Hamburg, Germany) was placed in the third ventricle as described elsewhere [14]. In brief, a 7cm midline skin incision over the frontal and parietal bones was made and the sagittal and coronary sutures were exposed. A manually operated twist drill was used to place the burr hole 1cm lateral to the sagittal suture at the upper part of the frontal bone. The dura was perforated and the bolt was fixed with its trajectory aiming towards the midline at an angle of 90˚ against the frontal bone. Then, the EVD catheter was carefully inserted until CSF could be extracted. Insertion depth was typically 6.5 to 7 cm. For continuous ICP measurement an ICP monitor (Spiegelberg GmbH&Co.KG; Hamburg, Germany) was used. The ventricular drain was connected to an external drainage and monitoring system (Duet External Drainage and Monitoring System®, Medtronic Inc., Minneapolis, Minnesota, USA). In order to ascertain stable ICP levels the EVD catheter was connected to a surge chamber filled with normal saline. By adjusting the height of this surge chamber the respective ICP levels were obtained.

## Ultrasound measurements

Ultrasound examinations were performed using a Philips iU22 ultrasound machine (Philips healthcare, Andover, MA, USA) with a 7-17MHz linear array transducer (L15-7io). The mechanical index was set at 0.3 [14–16]. Ultrasound loops were recorded at baseline and 10 minutes after achieving stable ICP target values of 20, 30, 40, 50 and 60 mm Hg. At each ICP step, three consecutive ultrasound examinations were conducted. Transbulbar sonography was only performed on the right side since all pigs were positioned in a left lateral decubitus position. All ultrasound loops were analysed offline by two investigators who were blinded to the intracranial pressure level and to the animal. As per convention, the diameters of interest (optic nerve and optic nerve sheath) were measured 3 mm behind the papilla [2, 3]. The average of the means of the three consecutive measurements from each investigator was used. The diameter of the perineural space (PNS) was calculated by subtracting the diameters of ON from ONS.

The areas of ON and ONS at baseline and at ICP values of 60 mmHg were calculated using the formula area = $\pi$ / diameter $^2$. The area of the perineural space was calculated by subtracting the optic nerve area from the optic nerve sheath area.

## Statistical analysis

Despite normal data distribution (Shapiro-Wilk-Test), values are displayed as median, range and 95% confidence intervals for better interpretation of data distribution. The effect of increasing intracranial pressure on the different diameters was assessed using Friedman Test. In order to determine at which ICP the diameter became significantly different compared to baseline, each consecutive ICP value was compared to baseline using Dunn's test for multiple comparisons. Regression analysis were used for identifying the relationship between intracranial pressure and ultrasound assessments. Intra- and inter-investigator comparisons were made using coefficients of variation. ANOVA for repeated measurements with two within-group factors (ICP level and investigator) was used to assess a possible interaction of investigator and diameter with increasing ICP levels. Standard statistical software package was used for analysis of data (GraphPad Prism 8, GraphPad Software, USA).

## Results

Placement of the external ventricular drain catheter was successful in all eight animals. Median baseline ICP was 8 mmHg (range 6 to 12 mmHg). Hemodynamic characteristics for each ICP level are presented in Table 1. The sonographic measurements could be obtained for every ICP level in all animals. The inter-investigator coefficient of variation was 2.3% (range 0% - 9.4%). The intra-investigator coefficient of variation was 8% (range 0% - 21%). Details are presented in Table 2.

The diameters of the ON (baseline diameters 95% CI 0.33 cm to 0.45 cm to diameters at ICP 60 mmHg 95% CI 0.57 cm to 0.82 cm) and ONS (baseline diameters 95% CI 0.79 cm to 0.98 cm to diameters at ICP 60mmHg 95% CI 1.02cm to 1.38cm) increased significantly with increasing ICP level (p <0.002, full range of values Fig 2). However the diameter of the perineural space (baseline diameter 95% CI 0.40 cm to 0.59 cm to diameters at ICP 60 95% CI 0.38 cm to 0.62 cm) did not change significantly (p = 0.399). Multiple comparisons allowed differentiation between baseline (95% CI 0.34 cm to 0.45 cm) and values $\geq$40 mmHg for ON (95% CI 0.51cm to 0.66cm; p = 0.017) and between baseline (95% CI 0.79 cm to 0.98 cm) and values $\geq$50 mmHg for ONS (95% CI 1.02 cm to 1.34 cm; p = 0.006). The increase of ICP was paralleled by a linear increase of diameters of both ON ($R^2$ = 0.513, p<0.0001) and ONS (p $R^2$ = 0.364, <0.0001) however not for the PNS ($R^2$ = 0.047, p = 0.152).

ANOVA for repeated measurements showed a significant effect of ICP level on OND (p = 0.031) but neither a significant effect of investigator (p = 0.063), nor an interaction between the two (p = 0.089).

The median optic nerve area increased from 10.2 mm$^2$ at baseline to 36.3 mm$^2$ at ICP level of 60mmHg. The median optic nerve sheath area increased from 60.8 mm$^2$ at baseline to 120.8 mm$^2$ at ICP level of 60mmHg. The calculated perineural space area increased from 50.6 mm$^2$ at baseline to 84.5 mm$^2$ at ICP level of 60mmHg.

**Table 1. Cardiopulmonary characteristics.**

| Stage ICP | Baseline | 20 | 30 | 40 | 50 | 60 | Friedmann Test |
|---|---|---|---|---|---|---|---|
| **MAP (mmHg)** | 82 [79–108] | 82 [71–105] | 89 [72–122] | 105 [76–158] | 110 [79–179] | 156 [79–203] | p = 0.006 |
| **Heart rate (beats/min)** | 88 [76–123] | 101 [94–132] | 97 [78–155] | 126 [81–156] | 118 [79–206] | 137 [83–250] | p = 0.172 |
| **CPP (mmHg)** | 75 [72–103] | 62 [51–85] | 59 [42–92] | 65 [36–118] | 60 [29–129] | 98 [19–143] | p = 0.255 |
| **pCO2 (mmHg)** | 39 [35–41] | 38 [34–41] | 38 [37–39] | 38 [36–40] | 38 [37–40] | 40 [38–44] | p = 0.258 |

MAP = mean arterial pressure. CPP = cerebral perfusion pressure. Values are given as median and range.

**Table 2. Inter- and intra-investigator coefficients of variation.**

|  | optic nerve diameter | optic nerve sheath diameter | all structures |
|---|---|---|---|
| Inter-investigator coefficient of variation overall | 2.7% [0.3–9.4%] | 2.2% [0–8.6%] | 2.3% [0–9.4%] |
| Baseline ICP level | 3.0% [0.3–7.7%] | 1.5% [0–6%] | 2.2% [0–7.7%] |
| ICP level 20mmHg | 4.5% [0.9–9.4%] | 2.9% [1.3–5.7%] | 4.0% [0.9–9.4%] |
| ICP level 30mmHg | 2.4% [1.3–8.1%] | 1.9% [0.1–5.2%] | 2.1% [0.1–8.1%] |
| ICP level 40mmHg | 3.6% [1.3–6.3%] | 7.7% [0.3–8.6%] | 3.8% [0.3–8.6%] |
| ICP level 50mmHg | 1.8% [1.1–7.8%] | 2.8% [0.1–8.3%] | 2.1% [0.1–8.3%] |
| ICP level 60mmHg | 1.3% [0.3–6.3%] | 2.4% [0.2–7.7%] | 1.7% [0.2–7.7%] |
| Intra-investigator coefficient of variation overall | 10% [1–21%] | 5% [0–20%] | 8% [0–21%] |
| Baseline ICP level | 8.5% [3–21%] | 7.5% [0–15%] | 7.5% [0–21%] |
| ICP level 20mmHg | 10.5% [2–18%] | 5.5% [2–14%] | 7.0% [2–18%] |
| ICP level 30mmHg | 9.5% [1–17%] | 4.0% [1–14%] | 7.5% [1–17%] |
| ICP level 40mmHg | 7.5% [3–16%] | 3.5% [1–18%] | 5.0% [1–18%] |
| ICP level 50mmHg | 7.0% [1–17%] | 5.0% [0–15%] | 7.0% [0–17%] |
| ICP level 60mmHg | 5.5% [0–21%] | 6.0% [1–20%] | 5.5% [0–21%] |

ICP = intracranial pressure. All structures = optic nerve diameter and optic nerve sheath diameter. Values are given as median and [range]

## Discussion

This porcine study confirms that ultrasound measurements of the ONS diameter increase and correlate with increasing ICP. This is in line with results published in humans and pigs [2, 3, 9].

When analyzing the two different components of the ONS diameter–the optic nerve and the perineural space diameter–ONS diameter expansion can exclusively be attributed to an increase in ON diameter while the perineural space diameter shows no significant expansion. The finding that optic nerve diameter increased to a larger extent as the CSF filled perineural space diameter is unexpected. The following factors could have attributed to this finding: Capillaries and venules within the ON may have become congested. Of note, pigs have twice as many capillaries within ON compared to humans [17]. While mean arterial pressure significantly increased with increasing ICP, CPP remained unchanged up to ICP values of 50 mmHg, suggesting that cerebral perfusion probably did not increase with increasing ICP. In contrast, there is evidence that cerebral perfusion is limited by cerebral venous outflow when CPP is below 60–70 mmHg [18]. In our experiment, median CPP values were between 59 and 65 mmHg with ICP values between 20 and 50 mmHg. It is therefore conceivable that venous congestion contributed to the increase in ON diameter. Further, increasing ICP may amplify the effect of venostasis on optic nerve swelling. Diminished cerebral venous return can also be partially explained by the fact that head and the upper part of the bodies were not elevated in our pig model as opposed to standard clinical practice with head elevation in patients experiencing elevated intracranial pressure.

So far most published literature consider CSF shifts as a change detectable by ultrasound and therefore the focus is on optic nerve sheath diameter without measuring optic nerve diameter[3, 19].

However optic nerve diameter measurements have been assessed and validated. A high intra- and interobserver reliability was documented for OND measurements in healthy subjects[20]. In patients with multiple sclerosis OND can depict optic nerve atrophy[21]. Two studies conducted in patients with idiopathic intracranial hypertension found no significant difference in optic nerve diameter between control groups and patients with elevated

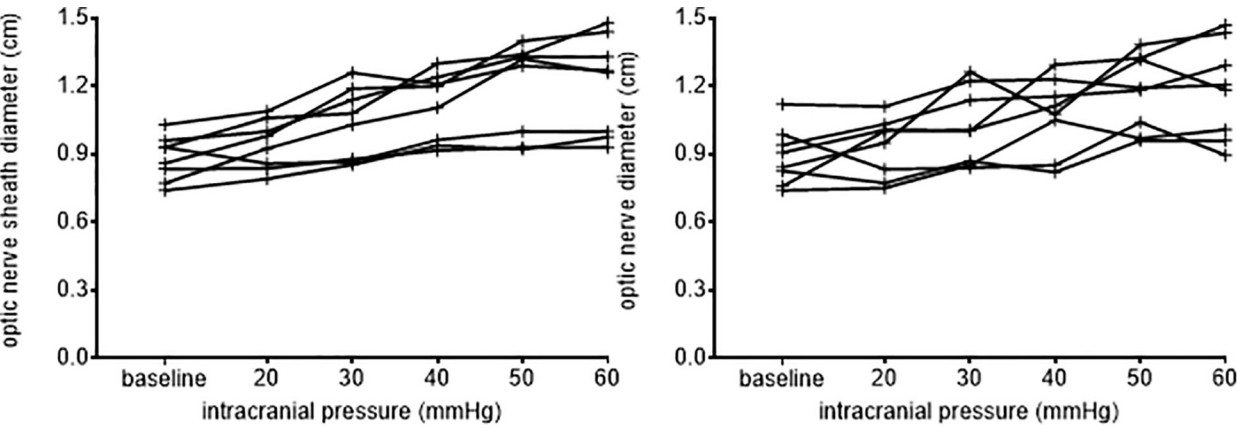

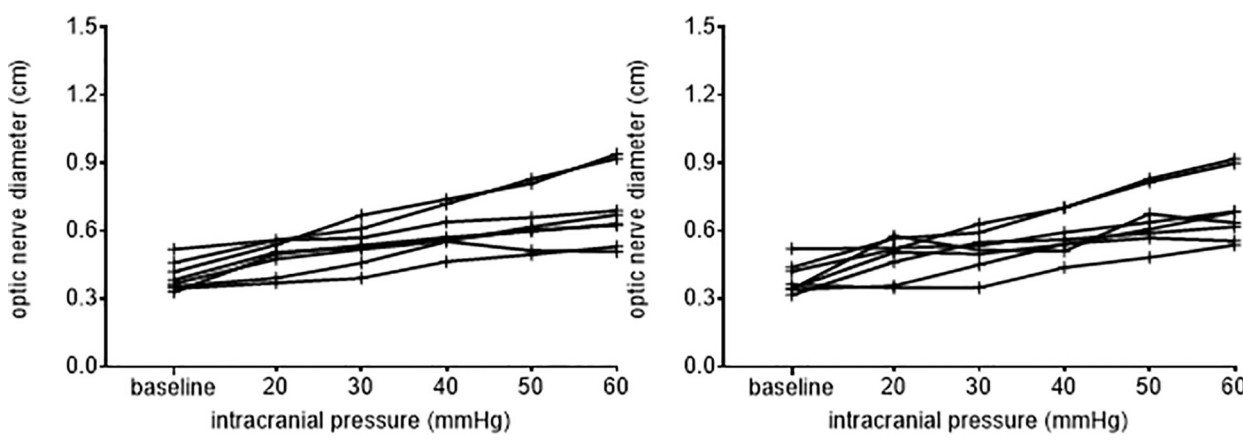

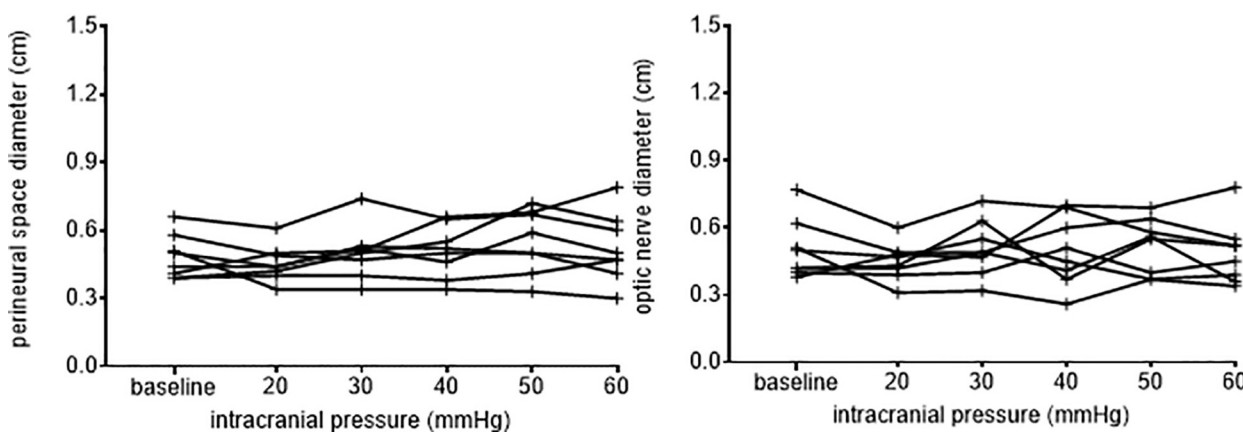

**Fig 2. Individual values of all eight pigs concerning optic nerve diameter, optic nerve sheath diameter and perineural space diameter for the respective ICP levels.** Left row illustrates measurements from investigator 1. Right row measurements from investigator 2.

intracranial pressure[22, 23]. However intracranial pressure levels in these studies were mildly to maximum moderately increased (ICP levels between 18 and 25 mmHg). Given the much smaller diameter of optic nerve compared with optic nerve sheath, differences might have been too small to be detected. We acknowledge, that in our study the increases in ON diameter became statistically significant only at ICP values of ≥40 mmHg. The study from Topcuoglo M et al measured different diameters within the orbit in brain death and non brain death patients[1]. Optic nerve diameter was significantly larger (p < 0.001) in brain death patients compared to non brain death patients. This finding might support our finding that also non cerebrospinal fluid changes attributed to elevated intracranial pressure can be detected by measuring optic nerve diameter using transbulbar ultrasound.

Since all measurements were obtained within minutes after the establishment of a new ICP level, the adaptive processes in cerebrospinal fluid and blood circulation may not have occurred yet. This might be particularly true for the cerebrospinal fluid circulation with its slow and steady production [24]. Despite stable perineural space diameter the area of the peri-neural space still increases due to the increase of inner and outer circumference of the peri-neural space.

Our study and therefore our findings have clear limitations. First it is an animal study with inherent problems in transmissibility to humans. The differences in optic nerve vascularization might have significantly influenced our findings. Our setup investigated only short term changes, and there is an ongoing debate about what ICP model is the most accurate one. Many different models with different advantages and disadvantaged have been proposed such as infusion of artificial CSF through a cannula placed in the subdural/subarachnoid space [25, 26], insertion and inflation of a balloon catheter into the ventricle system [26] or in the sub-dural space [27, 28], infusion of saline through an intra-parenchymal catheter [28, 29] and infusion of Hartmann solution in the lumbar subarachnoid space [29, 30] or reduction of cere-bral venous return [9, 30]. Although we confirmed the linear increase of ONS diameter with ICP, our method was not able to indicate threshold values for increases in ICP below 40 mmHg, which limits the usefulness of the method in clinical practice.

One strength of our study is the fact that we were well able to control and obtain stable ICP levels and that we performed measurements over a very wide ICP range including very high ICP levels. Further, the investigators were blinded to the respective ICP levels when analyzing the ultrasound loops. The reasonably small coefficients of variation for inter-investigators indicate that the method used is accurate [7, 31].

Our findings should be confirmed in humans with increased ICP.

## Conclusion

In a porcine ICP model, the diameter of the optic nerve correlates linearly with ICP and can be reliably measured using transbulbar ultrasonography. The increase of the ONS diameter can be mainly attributed to an increase of the ON. In contrast, the diameter of the perineural space does not correlate with ICP.

## Supporting information

**S1 File.**
(XLSX)

## Acknowledgments

The study was conducted at the Experimental Surgery Unit of the Department of Clinical Research, University of Bern, Bern, Switzerland.

The authors would like to thank Olgica Beslac and Daniel Mettler from the Experimental Surgery Unit and Kay Nettelbeck from the Department of Intensive Care Medicine, Inselspital, Bern University Hospital and University of Bern, Switzerland for their skillful support.

## Author Contributions

**Conceptualization:** M. Glas, C. Fung, S. M. Jakob, J. Beck, W. J. Z'Graggen, Andreas Bloch.

**Formal analysis:** I. Zubak, A. Schuetz, S. M. Jakob, J. Beck, W. J. Z'Graggen, Andreas Bloch.

**Funding acquisition:** S. M. Jakob, Andreas Bloch.

**Investigation:** R. Mija, I. Zubak, M. Glas, C. Fung, W. J. Z'Graggen.

**Methodology:** R. Mija, I. Zubak, M. Glas, C. Fung, S. M. Jakob, J. Beck, W. J. Z'Graggen, Andreas Bloch.

**Project administration:** S. M. Jakob, W. J. Z'Graggen.

**Resources:** S. M. Jakob, W. J. Z'Graggen.

**Supervision:** S. M. Jakob, W. J. Z'Graggen.

**Writing – original draft:** R. Mija, A. Schuetz, S. M. Jakob, W. J. Z'Graggen, Andreas Bloch.

**Writing – review & editing:** R. Mija, A. Schuetz, M. Glas, C. Fung, S. M. Jakob, J. Beck, W. J. Z'Graggen, Andreas Bloch.

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
