## [Decision Letter · Decision Letter 0]

6 Sep 2019

PONE-D-19-21626

Correlation of optic nerve and optic nerve sheath diameter with intracranial pressure in pigs

PLOS ONE

Dear Dr. Bloch,

Thank you for submitting your manuscript to PLOS ONE. After careful consideration, we feel that it has merit but does not fully meet PLOS ONE’s publication criteria as it currently stands. Therefore, we invite you to submit a revised version of the manuscript that addresses the points raised during the review process.

We would appreciate receiving your revised manuscript by Oct 21 2019 11:59PM. To enhance the reproducibility of your results, we recommend that if applicable you deposit your laboratory protocols in protocols.io, where a protocol can be assigned its own identifier (DOI) such that it can be cited independently in the future. For instructions see: http://journals.plos.org/plosone/s/submission-guidelines#loc-laboratory-protocols

We look forward to receiving your revised manuscript.

Kind regards,

Nimesh Patel

Academic Editor

PLOS ONE

Journal Requirements:

1. To comply with PLOS ONE submissions requirements, please provide methods of sacrifice in the Methods section of your manuscript.

Additional Editor Comments (if provided):

The work presented is important for understanding intracranial pressure. However, as pointed out by both expert reviewers, there is need to accurately describe the methodology used. Please address this major concern, along with other comments the reviewers have made.

Reviewers' comments:

Reviewer's Responses to Questions

**Comments to the Author**

1. Is the manuscript technically sound, and do the data support the conclusions?

Reviewer #1: Yes

Reviewer #2: Partly

2. Has the statistical analysis been performed appropriately and rigorously? 

Reviewer #1: Yes

Reviewer #2: I Don't Know

3. Have the authors made all data underlying the findings in their manuscript fully available?

Reviewer #1: Yes

Reviewer #2: No

4. Is the manuscript presented in an intelligible fashion and written in standard English?

Reviewer #1: No

Reviewer #2: Yes

5. Review Comments to the Author

Reviewer #1: Dear Authors,

I enjoyed reviewing your interesting and novel work. The finding that the optic nerve diameter changes with changes in ICP is intriguing. I would like the following issues addressed that will help make this manuscript more compelling.

1. Given the known technical challenges with visualizing optic nerve anatomy in humans, it is necessary for including a figure showing the actual ultrasound images of the relevant anatomy, not just a diagram. This will give the reader the important information regarding the technical aspects of the ultrasonic technique.

2. Throughout the manuscript, please include 95% CIs whenever the p-values are reported, this will give one a better idea of the precision of the results.

3. In the introduction, pg 3, line 65. "capillaries may become congested" I am not sure I understand why the capillaries would be selectively congested, and not the venules, please clarify, and correct if indicated

4. Page 5, line 102, "left decubitus position" Do you mean, "left lateral decubitus position"?

5. Please include a picture of the experimental setup if possible for illustration

6. What methods were employed to ensure the ICP readings were accurate? Was there any calibration performed. Were the ICP waveforms evaluated for responsiveness to physiological changes in pulse and respiration to determine the accuracy and patency of the transduction system?

7. Page 10 , line 1, "perineural space" I believe this is an error and should be "optic nerve sheath" please check.

8. The discussion is inadequate with regards to the measurement of optic nerve diameter. Please elaborate on why this structure has not been measured before, or whether it has in humans and or animals in relationship to the ICP, or even as an anatomical measurement with ultrasound. Consider incorporating the article, "Chen, Q et al High-resolution transbulbar ultrasonography helping differentiate intracranial hypertension in bilateral optic disc oedema patients. Acta Ophthalmologica. 2017

9. page 9, "Further, elevated ICP levels cause additional vasogenic edema..." I do not understand the physiological mechanism referred to here. Vasogenic edema does not accompany all ICP elevation, especially if the ICP elevation does not disrupt the blood brain barrier, consider omitted this statement or clarifying it.

10.

Please correct multiple typographical errors for example...

pg 4, line 83, "intracompartimental"

Pg 9, line 8, "extend" should be "extent"

pg 9, "liquor" is usually referred to cerebrospinal fluid, please correct

Reviewer #2: Thanks for the opportunity to review this article. Since the entire work is relying on ultrasound image-based measurement values, it is imperative that representative images are shown, along with the measurement points. This is even more important since 40 kg porcine optic nerve at baseline should be smaller (human ON rarely exceeds 3 mm). The diagram (Figure 1) is clearly insufficient.

Porcine optic nerve imaging is very difficult due to a number of impediments, including positional instability of the eyes, near-impossibility to bypass the large lens, limited probe positioning options, and others. These difficulties, combined with unexpected results, make it even more important for the reader to see the actual images of the measured structures with the measuring technique.

6. PLOS authors have the option to publish the peer review history of their article (what does this mean?). If published, this will include your full peer review and any attached files.

Reviewer #1: Yes: Eric Bershad

Reviewer #2: No

---

## [Author Response · Author response to Decision Letter 0]

23 Oct 2019

Editor comments

To comply with PLOS ONE submissions requirements, please provide methods of sac-rifice in the Methods section of your manuscript.

Response

In order to comply with PLOS ONE submissions requirements we have added details concerning sacrifice of the pigs in the methods section. The following sentence was added. “… At the end of the experiment a bolus of 40 mmol potassium chloride was given to sacrifice the animal.” 

Reviewer #1: 

Dear Authors,

I enjoyed reviewing your interesting and novel work. The finding that the optic nerve diameter changes with changes in ICP is intriguing. I would like the following issues addressed that will help make this manuscript more compelling.

Reviewer comment

1. Given the known technical challenges with visualizing optic nerve anatomy in hu-mans, it is necessary for including a figure showing the actual ultrasound images of the relevant anatomy, not just a diagram. This will give the reader the important infor-mation regarding the technical aspects of the ultrasonic technique.

Response

We have included the following figure showing two actual ultrasound images with the relevant anatomy marked with arrows and stars. One image was recorded at baseline ICP (9 mmHg) and the second one at an ICP of 40mmHg. We have therefore excluded the previous schematic figure 1. This will hopefully improve the illustration of the tech-nical aspects.

Fig 1: The upper transbulbar ultrasound image was recorded at baseline ICP – the lower image was recorded at ICP of 40mmHg. All measurements were obtained 3 mm behind the papilla as indicated by the two stars connected with the dotted line. Optic nerve diameter increased from 0.36 cm to 0.48 cm as illustrated by the black arrow. Op-tic nerve sheath diameter increased from 0.79 cm to 0.86 cm as illustrated by the white arrow.

Reviewer comment

2. Throughout the manuscript, please include 95% CIs whenever the p-values are re-ported, this will give one a better idea of the precision of the results.

Response

We have followed your valid input and we have added the respective 95% confidence intervals when p- values were reported.

Reviewer comment

3. In the introduction, pg 3, line 65. "capillaries may become congested" I am not sure I understand why the capillaries would be selectively congested, and not the venules, please clarify, and correct if indicated

Response

Thanks to your input we have corrected the obviously physiological inprecise state-ment “capillaries may become congested” to “capillaries and venules within ON may become congested…”

Reviewer comment

4. Page 5, line 102, "left decubitus position" Do you mean, "left lateral decubitus posi-tion"?

Reponse 

We have revised the term “left decubitus position” to “left lateral decubitus position”.

Reviewer comment

5. Please include a picture of the experimental setup if possible for illustration

Response

We did not take any picture during the trial so we cannot accommodate this demand. 

Reviewer comment

6. What methods were employed to ensure the ICP readings were accurate? Was there any calibration performed. Were the ICP waveforms evaluated for responsiveness to physiological changes in pulse and respiration to determine the accuracy and patency of the transduction system?

Response

For ICP-monitoring a bolt-kit silver-bearing EVD catheter system (Silverline-Bolt-Kit-catheter; Spiegelberg GmbH & Co. KG; Hamburg, Germany) was implanted. This cathe-ter type is equipped with an intraparenchymal/intraventricular ICP measurement sys-tem via an air-pouch mounted close to the tip of the drainage catheter. The air-pouch is connected via a separate lumen to a standalone ICP monitor. This monitor itself is connected to the multi-modular monitor (S/5 Critical Care Monitor®; Datex-Ohmeda, GE Healthcare, Helsinki, Finland) where the ICP waveform is displayed. The Spiegel-berg ICP measurement system has an automatic calibration algorhythm and a ne-glectable drift over time. In addition, conventional ICP measurement is also possible via the drainage catheter. After implantation of the EVD catheter, the ICP measured with the Spiegelberg monitor was compared with the conventional ICP measurement via the CSF coupled pressure transducer to check for accuracy. Furthermore, the ICP waveform was evaluated for responsiveness to manoevers reducing the venous back flow. For the further experimental procedures, the Spiegelberg ICP monitoring system was used.

Reviewer comment

7. Page 10 , line 1, "perineural space" I believe this is an error and should be "optic nerve sheath" please check.

Response

Given that the effective area (see calculations below) of the perineural space increas-es, the statement is correct. However in order to clarify and specify this we have changed the sentence from “Despite stable perineural space diameter the area of the perineural space still increases due to the overall increase of the combined diameters of the ON and perineural space” to “Despite stable perineural space diameter the area of the perineural space still increases due to the increase of inner and outer circumfer-ence of the perineural space” . 

Calculations: 

Calculating the areas of the components ON and ONS from baseline to values at 60 mmHg using the formula area = π / diameter 2 results in the following values:

Area ON (optic nerve) at baseline = 10.2 mm2

Area ON (optic nerve) at ICP 60 mmHg = 36.3 mm2

Area ONS (optic nerve sheath) at baseline = 60.8 mm2

Area ONS (optic nerve sheath) at 60 mmHg = 120.8 mm2

The area of the perineural space can be calculated by subtracting the optic nerve area from the optic nerve sheath area. This results in the following values:

Area PNS (perineural space) at baseline = 50.6 mm2

Area PNS (perineural space) at 60 mmHg = 84.5 mm2

We have further tried to emphasize this finding by integrating the calculation and the results of the respective changes of the different areas into the manuscript. 

Reviewer comment

8. The discussion is inadequate with regards to the measurement of optic nerve diame-ter. Please elaborate on why this structure has not been measured before, or whether it has in humans and or animals in relationship to the ICP, or even as an anatomical measurement with ultrasound. Consider incorporating the article, "Chen, Q et al High-resolution transbulbar ultrasonography helping differentiate intracranial hypertension in bilateral optic disc oedema patients. Acta Ophthalmologica. 2017

Response

Thank you for providing this interesting reference article. It is in fact difficult to find op-tic nerve diameter measurements in articles concerning transbulbar sonography. It is somewhat hard to elaborate the reasons for that. The most obvious explanation might be that no difference in optic nerve diameter was found which was also the case in the paper from “Chen et al”. Most studies on transbulbar sonography were conducted in patients with mild to maximally moderately increased ICP levels. Given the much smaller diameter of the optic nerve compared with the optic nerve sheath differences might be missed in these ICP ranges. In the present study, the increase in ON was significant only with ICP levels ≥40 mmHg Further, most authors only considered the cerebrospinal fluid shifts as a change detectable by ultrasound and they therefore fo-cused on optic nerve sheath diameter changes only. 

The article from “Topcuoglo M et al, Transorbital Ultrasonographic Measurements of Optic Nerve Sheath Diameter in Brain Death. Journal Neuroimaging. 20151” measured optic nerve diameter and optic nerve sheath diameter in brain-death and non brain-death patients. Optic nerve diameter was significantly larger (p < 0.001) in brain-death patients compared to non brain-death patients. This finding might support our finding that also non cerebrospinal fluid changes attributed to elevated intracranial pressure can be detected by transbulbar ultrasound by measuring optic nerve diameter. 

We have adjusted the discussion part of the manuscript. The abstract related to your comment reads as follows: 

So far most published literature consider CSF shifts as a change detectable by ultra-sound and therefore the focus is on optic nerve sheath diameter without measuring optic nerve diameter2,3. 

However optic nerve diameter measurements have been assessed and validated. A high intra- and interobserver reliability was documented for OND measurements in healthy subjects4. In patients with multiple sclerosis OND can depict optic nerve atro-phy5. Two studies conducted in patients with idiopathic intracranial hypertension found no significant difference in optic nerve diameter between control groups and patients with elevated intracranial pressure6,7. However intracranial pressure levels in these studies were mildly to maximum moderately increased (ICP levels between 18 and 25 mmHg). Given the much smaller diameter of optic nerve compared with optic nerve sheath, differences might have been too small to be detected. We acknowledge, that in our study the increases in ON diameter became statistically significant only at ICP values of ≥40 mmHg. The study from Topcuoglo M et al measured different diame-ters within the orbit in brain death and non brain death patients1. Optic nerve diameter was significantly larger (p < 0.001) in brain death patients compared to non brain death patients. This finding might support our finding that also non cerebrospinal fluid changes attributed to elevated intracranial pressure can be detected by measuring op-tic nerve diameter using transbulbar ultrasound. 

Reviewer comment

9. page 9, "Further, elevated ICP levels cause additional vasogenic edema..." I do not understand the physiological mechanism referred to here. Vasogenic edema does not accompany all ICP elevation, especially if the ICP elevation does not disrupt the blood brain barrier, consider omitted this statement or clarifying it.

Response

We have omitted this statement due to your valid comment. Thank you. 

Reviewer comment

10.

Please correct multiple typographical errors for example...

pg 4, line 83, "intracompartimental"

Pg 9, line 8, "extend" should be "extent"

pg 9, "liquor" is usually referred to cerebrospinal fluid, please correct

Response

We have thoroughly reviewed the manuscript with regards to typographical errors. We thankfully integrated your corrections.

Reviewer #2: 

Reviewer comment

Reviewer #2: Thanks for the opportunity to review this article. Since the entire work is relying on ultrasound image-based measurement values, it is imperative that repre-sentative images are shown, along with the measurement points. This is even more important since 40 kg porcine optic nerve at baseline should be smaller (human ON rarely exceeds 3 mm). The diagram (Figure 1) is clearly insufficient.

Porcine optic nerve imaging is very difficult due to a number of impediments, including positional instability of the eyes, near-impossibility to bypass the large lens, limited probe positioning options, and others. These difficulties, combined with unexpected results, make it even more important for the reader to see the actual images of the measured structures with the measuring technique.

Response

Thank you for your valid comment. 

We have included the following figure showing two actual ultrasound images with the relevant anatomy marked with arrows and stars. One image was recorded at baseline ICP (9 mmHg) and the second one at an ICP of 40mmHg. We have therefore excluded the previous schematic figure 1. 

As pointed out in the discussion our findings and our setup have clear limitations in-cluding inherent problems in transmissibility to humans. The differences in optic nerve vascularization might have significantly influenced our findings. However our setup with stable left lateral decubitus position and deep sedation of the animals might have facilitated the difficulties that you had pointed out. Hopefully the current figure will help the reader to better understand and to be able to relate to the described method. 

Fig 1: The upper transbulbar ultrasound image was recorded at baseline ICP – the lower image was recorded at ICP of 40mmHg. All measurements were obtained 3 mm behind the papilla as indicated by the two stars connected with the dotted line. Optic nerve diameter increased from 0.36 cm to 0.48 cm as illustrated by the black arrow. Op-tic nerve sheath diameter increased from 0.79 cm to 0.86 cm as illustrated by the white arrow.

References

1. Topcuoglu MA, Arsava EM, Bas DF, Kozak HH. Transorbital Ultrasonographic Measurement of Optic Nerve Sheath Diameter in Brain Death. Journal of neuroimaging : official journal of the American Society of Neuroimaging. 2015;25(6):906-909.

2. Dubourg J, Javouhey E, Geeraerts T, Messerer M, Kassai B. Ultrasonography of optic nerve sheath diameter for detection of raised intracranial pressure: a systematic review and meta-analysis. Intensive care medicine. 2011;37(7):1059-1068.

3. Fernando SM, Tran A, Cheng W, et al. Diagnosis of elevated intracranial pressure in critically ill adults: systematic review and meta-analysis. Bmj. 2019;366:l4225.

4. Lochner P, Coppo L, Cantello R, et al. Intra- and interobserver reliability of transorbital sonographic assessment of the optic nerve sheath diameter and optic nerve diameter in healthy adults. Journal of ultrasound. 2016;19(1):41-45.

5. Candeliere Merlicco A, Gabaldon Torres L, Villaverde Gonzalez R, Fernandez Romero I, Aparicio Castro E, Lastres Arias MC. Transorbital ultrasonography for measuring optic nerve atrophy in multiple sclerosis. Acta neurologica Scandinavica. 2018;138(5):388-393.

6. Chen Q, Chen W, Wang M, et al. High-resolution transbulbar ultrasonography helping differentiate intracranial hypertension in bilateral optic disc oedema patients. Acta ophthalmologica. 2017;95(6):e481-e485.

7. Jeub M, Schlapakow E, Ratz M, et al. Sonographic assessment of the optic nerve and the central retinal artery in idiopathic intracranial hypertension. Journal of clinical neuroscience : official journal of the Neurosurgical Society of Australasia. 2019.

---

## [Decision Letter · Decision Letter 1]

11 Nov 2019

PONE-D-19-21626R1

Correlation of optic nerve and optic nerve sheath diameter with intracranial pressure in pigs

PLOS ONE

Dear Dr. Bloch,

Thank you for submitting your manuscript to PLOS ONE. After careful consideration, we feel that it has merit but does not fully meet PLOS ONE’s publication criteria as it currently stands. Therefore, we invite you to submit a revised version of the manuscript that addresses the points raised during the review process.

We would appreciate receiving your revised manuscript by Dec 26 2019 11:59PM. To enhance the reproducibility of your results, we recommend that if applicable you deposit your laboratory protocols in protocols.io, where a protocol can be assigned its own identifier (DOI) such that it can be cited independently in the future. For instructions see: http://journals.plos.org/plosone/s/submission-guidelines#loc-laboratory-protocols

We look forward to receiving your revised manuscript.

Kind regards,

Nimesh Patel

Academic Editor

PLOS ONE

Additional Editor Comments (if provided):

Thank you for your revisions. Please address carefully the requested change by the second reviewer.

Reviewers' comments:

Reviewer's Responses to Questions

**Comments to the Author**

1. If the authors have adequately addressed your comments raised in a previous round of review and you feel that this manuscript is now acceptable for publication, you may indicate that here to bypass the “Comments to the Author” section, enter your conflict of interest statement in the “Confidential to Editor” section, and submit your "Accept" recommendation.

Reviewer #1: All comments have been addressed

Reviewer #2: (No Response)

2. Is the manuscript technically sound, and do the data support the conclusions?

Reviewer #1: Yes

Reviewer #2: Partly

3. Has the statistical analysis been performed appropriately and rigorously? 

Reviewer #1: Yes

Reviewer #2: Yes

4. Have the authors made all data underlying the findings in their manuscript fully available?

Reviewer #1: Yes

Reviewer #2: No

5. Is the manuscript presented in an intelligible fashion and written in standard English?

Reviewer #1: Yes

Reviewer #2: Yes

6. Review Comments to the Author

Reviewer #1: (No Response)

Reviewer #2: Thanks for replacing a diagram with actual ultrasound images. However, these provided images are not much superior compared to the diagram. While a slight variation of the imaging plane is acceptable, I find it disappointing that only fragments of the images are shown. The probe that you are using is excellent for this purpose, and could probably allow bypassing the lens; Judging by the similarity and quality of the patterns in the two images, it seems like the lens was indeed bypassed (or at least, transected with a similar aberration). But the reader should not have to guess (or remain oblivious to the fact that the lens may introduce major distortions and affect the measurements of the posterior structures). The reader does not learn from these images, nor do these images serve to assure the reader of proper technique hence accurate results. I would strongly recommend providing full images with a depth scale, otherwise my concerns are not alleviated.

7. PLOS authors have the option to publish the peer review history of their article (what does this mean?). If published, this will include your full peer review and any attached files.

Reviewer #1: Yes: Eric Bershad

Reviewer #2: No

---

## [Author Response · Author response to Decision Letter 1]

25 Dec 2019

Editor comments

Thank you for your revisions. Please address carefully the requested change by the second reviewer.

Response

Thank you for the possibility to address this important issue recognised by the second reviewer. The figure 1 has been replaced by a new figure 1.

Reviewer #2: 

Thanks for replacing a diagram with actual ultrasound images. However, these provid-ed images are not much superior compared to the diagram. While a slight variation of the imaging plane is acceptable, I find it disappointing that only fragments of the imag-es are shown. The probe that you are using is excellent for this purpose, and could probably allow bypassing the lens; Judging by the similarity and quality of the patterns in the two images, it seems like the lens was indeed bypassed (or at least, transected with a similar aberration). But the reader should not have to guess (or remain oblivious to the fact that the lens may introduce major distortions and affect the measurements of the posterior structures). The reader does not learn from these images, nor do these images serve to assure the reader of proper technique hence accurate results. I would strongly recommend providing full images with a depth scale, otherwise my concerns are not alleviated.

Response

Thank you for this very valid comment. We have followed your recommendation and we have provided the full images with a depth scale. As you can see both images rep-resent images made from high definition zooms with a zoom box size of 2 cm – there-fore the depth scale does not start at 0. 

In both ultrasound images the lenses were indeed bypassed and therefore no major distorsions were caused by the lens nor were the measurements affected.

We have further supplemented the legend of the figure as follows:

The upper transbulbar ultrasound image was recorded at baseline ICP – the lower im-age was recorded at ICP of 40mmHg. All measurements were obtained 3 mm behind the papilla as indicated by the two stars connected with the dotted line. Optic nerve di-ameter increased from 0.36 cm to 0.48 cm as illustrated by the black arrow. Optic nerve sheath diameter increased from 0.79 cm to 0.86 cm as illustrated by the white arrow. 

In both ultrasound images the lens was bypassed in order to obtain the most accurate display of the posterior structures of the eye. Further both images were made by using a high definition zoom box of 2 cm size.

---

## [Decision Letter · Decision Letter 2]

13 Jan 2020

Correlation of optic nerve and optic nerve sheath diameter with intracranial pressure in pigs

PONE-D-19-21626R2

Dear Dr. Bloch,

We are pleased to inform you that your manuscript has been judged scientifically suitable for publication and will be formally accepted for publication once it complies with all outstanding technical requirements.

With kind regards,

Nimesh Patel

Academic Editor

PLOS ONE

Additional Editor Comments (optional):

Thank you for addressing all concerns of the reviewers.

Reviewers' comments:

Reviewer's Responses to Questions

**Comments to the Author**

1. If the authors have adequately addressed your comments raised in a previous round of review and you feel that this manuscript is now acceptable for publication, you may indicate that here to bypass the “Comments to the Author” section, enter your conflict of interest statement in the “Confidential to Editor” section, and submit your "Accept" recommendation.

Reviewer #1: All comments have been addressed

Reviewer #2: All comments have been addressed

2. Is the manuscript technically sound, and do the data support the conclusions?

Reviewer #1: Yes

Reviewer #2: Yes

3. Has the statistical analysis been performed appropriately and rigorously? 

Reviewer #1: Yes

Reviewer #2: Yes

4. Have the authors made all data underlying the findings in their manuscript fully available?

Reviewer #1: Yes

Reviewer #2: Yes

5. Is the manuscript presented in an intelligible fashion and written in standard English?

Reviewer #1: Yes

Reviewer #2: Yes

6. Review Comments to the Author

Reviewer #1: (No Response)

Reviewer #2: Thanks for clarifications and a detailed response to reviewer concerns. Congratulations with successful completion of a technically challenging experiment. Best wishes to the team.

7. PLOS authors have the option to publish the peer review history of their article (what does this mean?). If published, this will include your full peer review and any attached files.

Reviewer #1: Yes: Eric Bershad

Reviewer #2: No

---

## [Editor Report · Acceptance letter]

21 Jan 2020

PONE-D-19-21626R2 

Correlation of optic nerve and optic nerve sheath diameter with intracranial pressure in pigs 

Dear Dr. Bloch:

I am pleased to inform you that your manuscript has been deemed suitable for publication in PLOS ONE. Congratulations! Your manuscript is now with our production department. 

With kind regards,

on behalf of

Dr. Nimesh Patel 

Academic Editor

PLOS ONE